# Immunoceuticals: Harnessing Their Immunomodulatory Potential to Promote Health and Wellness

**DOI:** 10.3390/nu14194075

**Published:** 2022-09-30

**Authors:** Sophie Tieu, Armen Charchoglyan, Lauri Wagter-Lesperance, Khalil Karimi, Byram W. Bridle, Niel A. Karrow, Bonnie A. Mallard

**Affiliations:** 1Department of Animal Biosciences, University of Guelph, Guelph, ON N1G 2W1, Canada; 2ImmunoCeutica Inc., Cambridge, ON N1T 1N6, Canada; 3Advanced Analysis Centre, University of Guelph, Guelph, ON N1G 2W1, Canada; 4Department of Pathobiology, University of Guelph, Guelph, ON N1G 2W1, Canada

**Keywords:** immunoceuticals, nutraceuticals, immunomodulation, immunocompetency

## Abstract

Knowledge that certain nutraceuticals can modulate the immune system is not new. These naturally occurring compounds are known as immunoceuticals, which is a novel term that refers to products and systems that naturally improve an individual’s immuno-competence. Examples of immunoceuticals include vitamin D3, mushroom glycans, flavonols, quercetin, omega-3 fatty acids, carotenoids, and micronutrients (e.g., zinc and selenium), to name a few. The immune system is a complex and highly intricate system comprising molecules, cells, tissues, and organs that are regulated by many different genetic and environmental factors. There are instances, such as pathological conditions, in which a normal immune response is suboptimal or inappropriate and thus augmentation or tuning of the immune response by immunoceuticals may be desired. With infectious diseases, cancers, autoimmune disorders, inflammatory conditions, and allergies on the rise in both humans and animals, the importance of the use of immunoceuticals to prevent, treat, or augment the treatment of these conditions is becoming more evident as a natural and often economical approach to support wellness. The global nutraceuticals market, which includes immunoceuticals, is a multi-billion-dollar industry, with a market size value of USD 454.55 billion in 2021, which is expected to reach USD 991.09 billion by 2030. This review will provide an overview of the immune system, the importance of immunomodulation, and defining and testing for immunocompetence, followed by a discussion of several key immunoceuticals with clinically proven and evidence-based immunomodulatory properties.

## 1. Introduction

The term “immunoceuticals” cannot be defined without first defining the term “nutraceuticals”, which was coined by Stephen L. DeFelice in 1989 as a blend of the two words “nutrition” and “pharmaceuticals” [1]. Despite the term nutraceuticals being first defined as “a food (or part of a food) that provides medical or health benefits, including the prevention and/or treatment of a disease”, there is currently no internationally standardized definition for “nutraceuticals”, and many have tailored or fine-tuned its definition overtime [1,2]. For example, Health Canada defined a nutraceutical as “a product isolated or purified from foods that is generally sold in medicinal forms not usually associated with food [3]. A nutraceutical is demonstrated to have a physiological benefit or provide protection against chronic disease”. For the purposes of this review, nutraceuticals are natural products that not only provide nutritional value but also help to support general health. When consumed above a threshold concentration, these natural products may have beneficial pharmacological effects. Therefore, the term nutraceuticals will be used to highlight the therapeutic values of certain natural products that are known to play a role in health performance beyond the nutritional value that they provide. Examples of nutraceuticals include, but are not limited to, certain vitamins, minerals, fatty acids, amino acids, peptides, proteins, and pre/probiotics.

“Immunoceuticals”, like nutraceuticals, is a portmanteau word that blends “immunity” and “pharmaceutical”. Therefore, any nutraceuticals that demonstrate beneficial immunomodulatory mechanisms that support an optimal immune system and/or modify immunological status to defend against various diseases such as cancers and infectious or autoimmune diseases can be separately categorized as immunoceuticals (Figure 1). Based on this definition, examples of immunoceuticals would include vitamin D3, mushroom-derived polysaccharides, plant-derived ergosterols, flavonols, terpenoids, carotenoids, aloe-associated polysaccharides, quercetin, omega-3 fatty acids, and micronutrients such as zinc and selenium, to name a few. Immunoceuticals are structurally different compounds with different modes of action. However, they all work to optimize immunological functions.

### 1.1. Brief Overview of the Immune System and Its Regulation

The immune system protects against invading pathogens such as bacteria, viruses, fungi, and parasites, as well as cancers, by distinguishing dangerous foreign and modified self-antigens from self-antigens from non-dangerous self and targets them for elimination using a diverse set of host defense mechanisms [4,5]. The immune system is composed of a collection of cells, molecules, tissues, and organs and contains two main branches: the innate and adaptive immune systems. Innate host defence, sometimes referred to as non-specific immunity provides the first line of defence using a wide array of cells and molecules against an invading pathogen. It is non-specific in that the innate defence mechanisms use a limited array of receptors to respond to dangerous molecules, and each response is mounted as if seeing them for the first time or with only transient forms of memory training. These mechanisms have broad specificity and are fast-acting, with responses initiated within seconds to hours.

The adaptive immune response can be further divided into two categories: antibody and cellular responses mediated by B- and T-lymphocytes, respectively. Adaptive immune responses are differentiated from innate responses in that they are antigen-dependent, clonally antigen-specific, and possess immunological memory that can last years, or even a lifetime. Due to its exquisite antigen specificity and durability, the adaptive immune defences, particularly upon initial exposure, require several days to two weeks to mount a response. However, as a result of immunological memory, subsequent exposure to the same antigen allows for a far more rapid and higher magnitude immune response that can usually eliminate an infection prior to the onset of the disease. It is also important to understand that the innate and adaptive immune systems are not independent, but rather communicate to provide a carefully orchestrated collection of defence mechanisms shaped by genetics of the host and the nature of the encountered invader.

When performing optimally, the innate and adaptive responses provide immunity, which means protection or exemption from disease, ideally accompanied by an inability to transmit the causative agent to others. Protection may be provided passively—for example, antibodies delivered to a newborn via the mother’s first milk, known as colostrum—or actively via naturally acquired immunity and artificially induced immunity. Naturally acquired immunity is commonly achieved following infection and recovery from exposure to a pathogen. Artificially acquired immunity may be achieved through vaccination, which attempts to recapitulate the gold-standard of naturally acquired immunity. However, it is critical to understand that not all immune responses are optimal or protective. The immune system is often referred to as a double-edged sword, where protection is carefully balanced against immuno-pathology or damage caused by prolonged overt or inappropriate immune responses. Thus, a critical role of immunoceuticals lies in helping to tune optimal host defence.

The regulatory mechanisms of the immune system are vast and complex, with many different regulators and factors involved that affect how the immune system functions and responds. It is widely recognized that the manifestation of infection varies greatly from mild to severe among individuals, and this is due, in part, to the genetics of both the host and the pathogen. In fact, around 5000 of the 23,000 genes in mammals are dedicated to host defence [6]. This makes sense, given that the immune system governs survival. However, adaptive changes induced by environmental influences, especially during early life development, also play a large role in helping to shape an individual’s immune system [7]. The interplay between genes and the environment is commonly referred to as gene-by-environmental interactions, and epigenetic mechanisms govern these. For example, environmental pathogen exposure helps shape the adaptive immune system repertoire in terms of what T- and B-cell clones predominate in a given host. The biology of the pathogen also plays a role; for example, its virulence, the portal of entry and the dose of exposure to the pathogen all affect disease outcome. Collectively, the host–pathogen–environment triad shapes disease consequences.

Epigenetic modifications are changes to DNA that do not involve alterations to the genomic sequence; typically involving DNA methylation, histone modifications, or small noncoding RNAs. These changes are induced by environmental stimuli, including but not limited to exposure to immunomodulatory immunoceuticals, such as colostrum [8], probiotics [9], quercetin [10], kaempferol [11], and curcumin [12]. Epigenetic modifications can occur in cells of the immune system and can be passed on with each cell division to daughter cells. Consequently, epigenetic modifications can potentially influence immunological phenotypes and impact disease outcomes. In some cases, these changes are passed on to future generations. For example, global and gene-specific DNA methylation and alterations in fatty acid content by arachidonic acid exposure are maintained across generations, which may prove beneficial to the innate immune system [13]. Additionally, the epigenetic effects of preconceptual paternal infection and activation of paternal immune responses can affect offspring phenotypes, particularly brain function, behavior and immunological functions across multiple generations without re-infection [14].

There are a variety of cells, such as regulatory T cells and dendritic cells, and molecules, such as transcription factors (e.g., nuclear factor-kB), cytokines, and microRNAs, that serve as key regulators of the immune system. Working in concert, this complex communication network ensures that the immune system is functioning effectively to distinguish between the normal self and the dangerous non-self and to ensure that the response is not excessive and potentially damaging to tissues. Continuous cross-talk between the innate and adaptive immune system, the immune system and the hypothalamic–pituitary–adrenal axis, the gut–brain connection, and the mucosal and central immune system are also important aspects of the regulation of the immune response. Cross-talk between innate and adaptive responses plays a key role in the regulation and appropriate activation of the adaptive immune system. For example, the activation of adaptive immunity is dependent on a subset of innate leukocytes called antigen-presenting cells, which activate helper T cells, which then go on to modulate the cellular and humoral immune responses of the adaptive immune system, which can also involve elements of the innate immune system (e.g., macrophages) [15]. Communication between the hypothalamic–pituitary–adrenal axis and the immune system, the gut–brain connection, as well as the common mucosal immune system can influence and regulate the function of leukocytes via the production of hormones secreted by the pituitary, adrenal, and other endocrine organs, as well as serving as an ideal environment to educate meningeal B cells to produce IgA antibodies against specific microbes, such as those causing meningitis [16,17]. The details of these connections are outside the scope of this review.

### 1.2. Inflammation and Immunopathology

Inflammation is an integral component of the immune system and is a protective strategy that is designed to alert the body to danger and recruit leukocytes to the site of infection or tissue trauma. Classical signs and symptoms of inflammation include heat, fever, redness, swelling, and pain, the combination of which can often lead to a loss of function in the affected tissue or organ. The molecular mechanism of inflammation is a complicated process that is tightly regulated by several key regulators, including inflammasomes [18], inflammatory caspases (e.g., caspases −1, −4, −5, −11, and −12) [19], and pro-resolving lipid mediators [20] to name a few. Cellular homeostasis must subsequently be restored after the harmful stimuli has been terminated to promote the healing of damaged tissues. However, chronic inflammation can occur if there is a failure in eliminating the noxious stimuli [21]. Genetic defects in the innate or adaptive host defence mechanisms can also result in the development of chronic inflammatory diseases, among other immunopathological disorders including autoimmunity, immunodeficiency, and hypersensitivity reactions (e.g., allergic reactions). Poorly regulated inflammatory responses and tissue damage resulting from inflammation are key immunopathological features [5]. Immunosenescence is another contributing factor due to an age-associated decline in immune responses, which contributes to increased risk of infectious and inflammatory diseases [22].

The importance of addressing chronic inflammation cannot be overstated, as inflammation plays a primary role in the etiology of many diseases including atherosclerosis, obesity, type 2 diabetes, asthma, inflammatory bowel disease, neurodegenerative diseases, rheumatoid arthritis, psoriasis and cancer [21]. Furthermore, our understanding of the pathological processes of chronic inflammatory diseases is relatively limited [21]. Developing safe and effective anti-inflammatory therapeutic interventions, including immunoceuticals, remains a crucial goal.

### 1.3. The Importance of Immunomodulation

Immunomodulation refers to all therapeutic interventions intended to beneficially modify the immune response [23]. Pathological conditions characterized by insufficient immunological function (immunodeficiency), or situations in which a normal immune response is suboptimal, overly aggressive, or inappropriate in controlling an infection or pathological condition, are circumstances in which augmentation or tuning of an immune response is desirable. Augmentation of the immune response in states of immunodeficiency is beneficial in preventing infection and fighting established infections and cancers [23]. Immunodeficiency can develop as a consequence of malnutrition, treatment of cancers (e.g., chemotherapy and radiation), viral infections (e.g., human immunodeficiency virus), and genetic defects or genetic predispositions [23,24].

Heritable and non-heritable factors in shaping human immune systems have been reported. Genetic variation is an important driver of immune variation, and factors such as age, sex, diet, environmental exposure, and microbiome are demonstrated to affect immune responses [25]. It has been well-established in animal models that individuals can be classified based on their ability to mount robust and balanced immune responses, into high, average or low immune responders [24,26,27]. The high immune responders are significantly less likely to become sick compared to the others and are, therefore, less likely to require therapeutic interventions [24,26]. Conversely, the average and low responders have a greater need for the benefits conferred by immunoceuticals which help to enhance their ability to make protective responses. The immune system is controlled by many genes whose expression levels differ from individual to individual, and this determines their propensity to mount protective immune responses following exposure to a pathogen. This is one reason why a diverse set of signs and disease outcomes are observed among individuals exposed to the same pathogen. There are also circumstances in which immunomodulation to attenuate a harmful immune response is desirable, and these circumstances include allergies and autoimmune diseases [23]. The potential to modify undesirable immune responses is a growing area of interest with vast potential to improve health. Genetic defects causing immunodeficiencies may require other interventions and are not within the scope of this review.

### 1.4. Traditional Immunomodulators

Traditionally, immunosuppressive drugs, immunomodulatory corticosteroids, vaccines, and antibiotics have been used to either augment immunological functions, or to suppress pathological immune responses. However, administration of these compounds can also be accompanied with negative consequences such as unwanted side effects, drug interactions and resistance, adverse events following vaccination, or antimicrobial resistance attributed to overuse of antibiotics. Additionally, most of these drugs end up in the urine and feces with poorly defined environmental consequences [28]. Furthermore, according to the United States Food and Drug Administration (FDA), up to 90% of all experimental drug compounds that go through clinical trials fail to gain FDA approval due to issues with efficacy, formulation, pharmacokinetics, toxicology, or clinical safety [29].

### 1.5. Clinical Benefits of Immunoceuticals

The importance of immunoceuticals to prevent, treat or augment treatment of pathological conditions such as cancers, viral and bacterial infections, chronic inflammatory diseases, autoimmune disorders and allergies is becoming more evident as a natural approach to support wellness [30]. In Figure 2, the potential utility of immunoceuticals is demonstrated by using infection with severe acute respiratory syndrome-coronavirus-2 (SARS-CoV-2) as an example, and/or adverse events following receipt of a vaccine intended to protect against.

For instance, the novel coronavirus disease that was first identified in 2019 (COVID-19). In Table 1, the applications of immunoceuticals are demonstrated in terms of treating respiratory and gastrointestinal tract infections, cancers, and acquired immunodeficiency syndrome (AIDS).

Immunoceuticals, as a specific subcategory of nutraceuticals, offer a focused approach of supporting the immune system and ensuring its optimal functioning. Many immunoceuticals could be made readily available with revised guidelines that focus on optimizing their immunomodulatory potential to facilitate minimizing disease burdens at the population level. One example is supplementing with higher concentrations of vitamin D than that traditionally recommended in nutritional guidelines.

### 1.6. Economic Considerations for Nutraceuticals

The global nutraceutical industry had a market size value of $454.55 billion USD in 2021. This is expected to reach $991.09 billion USD by 2030, expanding at an expected compound annual growth rate of 9.0% from 2021 to 2030 [46]. From the perspective of consumers, many immunoceuticals are available at relatively economical prices compared to pharmaceuticals. This is because, immunoceuticals are easily accessible as they are natural products generally found in foods or in the case of vitamin D by appropriate sun exposure. They are also supported by a wide breadth of scientific literature and clinical studies to support their safety, efficacy, and immunomodulatory properties [47].

### 1.7. Defining and Testing for Immunocompetence

Immunocompetence is defined as the ability of an organism’s immune system to elicit an appropriate immune response following exposure to an antigen [48]. Protection against pathogenic processes requires proper functioning immune system, which includes surveillance for antigens and the elicitation of a rapid and effective response following exposure that is resolved as quickly as possible. Since disturbances in immune functionality can affect immunocompetence thereby increasing an individual’s susceptibility to infection and disease, there is merit in measuring various immunological parameters as a way of assessing an individual’s immunocompetence.

Measuring leukocyte function and numbers is a way that immunocompetence can be assessed. Cells of the immune system are collectively referred to as leukocytes, which can be further divided into lymphocytes, monocytes, and granulocytes. Due to the diverse functional differences among these cells and their subsets, leukocyte function can be assessed using a number of functional assays [48,49]. Since not all leukocytes possess the ability to proliferate in response to antigen, migrate to the site of infection, possess phagocytic and cytotoxic properties, or generate antibodies, interferons and interleukins, particular assays are required to assess distinct cell functions. While many immune assays are performed *in vitro*, *in vivo* tests are also available and provide a more holistic assessment of immunocompetence. Examples of in vivo measures include the delayed-type hypersensitivity skin test, which is based on the reaction that occurs following an intradermal injection of an antigen and is an indicator of T cell responsiveness, or the concentration of antibodies produced following the administration of a vaccine or test antigen. Quantitative measures of the numbers of leukocytes and their products can also be used to assess immunocompetence instead of assessing cell activity. Total white blood cell counts (WBC) in humans only provide a crude quantitative measure of immunocompetence as the normal range for WBC values is extremely variable (5000–10,000 cells/mm^3^) [48]. Thus, deviations from the normal range can be difficult to interpret without a differential analysis of the absolute and relative numbers of lymphocytes and their subpopulations, monocytes, and granulocytes. Another quantitative assessment of immunocompetency includes assessing antibody status. For example, elevated concentrations of autoantibodies can be suggestive of autoimmune disease, whereas elevations of antibodies of the appropriate isotype against most pathogens may be indicative of protection, but antibodies against latent viruses such as the herpes simplex virus or Epstein-Barr virus may indicate reinfection or reactivation of the virus, and therefore, decreased immunocompetency [50]. As such, it is critical to understand the nature of the immune response being evaluated in order to provide accurate interpretation prior to making therapeutic recommendations.

### 1.8. Nutritional Immunology

Individual nutritional status is an important predictor of the health and effectiveness of a host’s immune system because the immune response against a pathogen, cancer or toxin is metabolically costly.

Undernutrition or imbalanced nutrition (malnutrition) are positively correlated with immunodeficiency, with almost all immunological effector mechanisms being affected, especially the non-specific defences and cell-mediated immunity [51]. Undernutrition has a deteriorative effect on immune responses, eventually resulting in the atrophy and dysfunction of immunological organs and tissues, and changes in the numbers, ratios, and functions of different leukocyte immunophenotypes [51]. Likewise, protein and energy malnutrition has been linked to destruction of cell-mediated immunity, phagocyte function and the complement system, and decreased antibody concentrations, especially secretory immunoglobulin A, and/or production of cytokines [52].

The past few decades have led to an increased understanding of how naturally occurring compounds, such as the immunoceuticals abscisic acid and conjugated linoleic acid; *n-*3 fatty acids; and vitamins A, D, and E can modulate immune responses. Moreover, the nutritional value that these naturally occurring compounds provide in the diet, they can also dynamically influence the immune system [51]. More than 65% of leukocytes in the body are found in the gut, thus making the gut one of the largest immunological organs [51]. Additionally, the innate immune receptors in the gut serve as primary targets for immunomodulation via the diet.

The key to preventing immunodeficiencies resulting from malnutrition or nutritional deficiencies is provision of whole and balanced nutrition starting right in the womb. Nutritional care is of utmost importance during pregnancy, infancy and childhood, and in old age, as these demographics are more vulnerable and have either an immune response bias, or an immature or aging immune system, respectively, that does not function at its optimal potential. A cost -effective approach to minimizing immunodeficiencies related to nutrition is to fortify foods with nutrients such as vitamin D, A, E, zinc and selenium, so they can be obtained without requiring changes to dietary habits, particularly when those lifestyle changes are not obtainable in certain scenarios [51]. Examples of foods that are fortified with such nutrients include cereals and cereal products, milk and milk products, fats and oils, beverages and infant formulas. Biofortification, such as the addition of probiotics, is another way in which foods can be fortified and act as immunoceuticals [53].

Micronutrient deficiencies in developed countries leading to immune system problems are mainly caused from dietary restriction (cultural or religious or personal habits), which may lead to the selection of certain foods and not others [51]. Therefore, consumption of a diverse diet is required to avoid nutritional deficiencies. Severe deficiency of nutrients (micro or macro) may require supplementation of nutrients along with a normal diet to improve immunity.

It is important to note that the supplementation of specific nutrients is recommended in situations of deficiency, and not in situations of sufficiency. Nutritional deficiency is defined as an inadequate supply of essential nutrients in the diet leading to malnutrition and/or disease [54]. In contrast, nutritional sufficiency is defined as an adequate supply of essential nutrients in the diet, thus preventing malnutrition and the development of diseases related to malnutrition. Supplementation of specific nutrients in situations of nutritional sufficiency may not always be beneficial and may cause harm. For example, supplementation of iron under disease challenge may not be beneficial as many pathogens rely on the bioavailability of iron to proliferate [55]. As such, host defence strategies have the ability to limit the availability of iron for bacteria by sequestering iron away [55]. Thus, the value of knowledge across immunological disciplines when dealing with immunoceuticals, including developmental and nutritional immunology and immunotoxicity, is of utmost importance. Certain immunoceuticals when not properly formulated, or when consumed at an inappropriate concentration can lead to immune system imbalances resulting in inflammation or immunosuppression.

## 2. Key Immunoceuticals and Their Immunomodulatory Properties

The immunomodulatory properties of several immunoceuticals are summarized in Table 2. The following sections will highlight a few of these key immunoceuticals.

### 2.1. Vitamin D and Immunomodulation

Vitamin D is not only a vitamin, but also a prohormone that acts as an immunomodulator and plays a vital role in maintaining calcium and bone homeostasis. It can be obtained from ultraviolet (UV) B-dependent endogenous production and/or from diet and supplements. Natural dietary sources of vitamin D include fatty fish such as salmon, mackerel, sardines and cod liver oil, and certain types of mushrooms, such as Shiitake, may contain relevant amounts of cholecalciferol or ergocalciferol (two major forms of vitamin D), especially if they are sun-dried [76]. In countries such as the United States and Canada, certain products such as dairy products may also be fortified with vitamin D, usually in the form of cow’s milk in the United States, and fluid milk (milk processed for beverage use) and margarine in Canada [77]. Therefore, vitamin D status between individuals will vary greatly depending on the extent of endogenous vitamin D production, which is partly influenced by genetics, latitude of residence, season, concentration of skin pigments and lifestyle (e.g., use of sunscreen and clothing choice), as well as their nutritional habits [78].

Vitamin D exists in two major forms, ergocalciferol (vitamin D2) and cholecalciferol (vitamin D3; 25-hydroxyvitamin D3 (25(OH)D3). Ergocalciferol is derived from the sterol ergosterol (previtamin D2) which is found in plants and fungi. In contrast, vitamin D3 is produced in the skin from 7-dehydrocholesterol when exposed to UVB radiation. Regardless of the form of vitamin D present, it must first be converted to calcitriol (1,25(OH)_2_D) before it can be biologically active and affect mineral metabolism and modulate the immune system [79]. After consumption, vitamin D is transported in the blood while bound primarily to the vitamin D binding protein (VDR), to the liver where it is converted to calcidiol (25(OH)D3) by the enzymes CYP2R1 and CYP27A1 [78]. Calcidiol is subsequently transported to the kidneys where it is converted into bioactive calcitriol by CYP27B1. Since calcidiol is the main circulating vitamin D metabolite, it is the most reliable parameter for determining vitamin D status in humans [80]. Although CYP27B1 is predominantly found in the kidneys, it has also been found in the placenta and leukocytes including monocytes, macrophages, dendritic cells (DCs) and B and T cells [81,82,83,84,85].

The expression of CYP27B1 by monocytes, macrophages and DCs is critically important since it allows them to convert inactive vitamin D into bioactive vitamin D [78]. Additionally, unlike renal CYP27B1, which is regulated by a negative feedback mechanism, CYP27B1 in monocytes, macrophages and DCs lacks a feedback mechanism, which allows these cells to produce high concentrations of calcitriol locally for immunomodulation (Figure 3) [78].

Vitamin D’s role in regulating calcium and phosphorus metabolism, as well as bone health has long been established. However, in recent decades, it has become evident that the functions of this vitamin also extend to the immune system, as demonstrated by the discovery of 1,25-dihydroxyvitamin D3 (1,25-(OH)_2_D3) receptors (VDR) in human blood-derived monocytes and active lymphocytes [86]. VDR expression has since been found in almost all cells of the immune system but are expressed at significantly higher concentrations in T-lymphocytes and macrophages, and even higher in developing T cells in the thymus, and established CD8+ cells [87,88]. The effect of vitamin D on different types of leukocytes differs greatly as control of VDR expression is based on the activation status of each type of leukocyte [88]. For example, expression of VDR in monocytes will decrease once they differentiate into macrophages or DCs. In contrast, upon activation, T cells will display a significantly higher concentration of VDR eight hours post-activation, reaching maximum concentration at 48 h [85,89]. Regardless of the different effects that vitamin D has on various leukocytes, a deficiency of vitamin D will result in insufficient and impaired innate and acquired responses, which consequently increases risk of infections [90].

#### 2.1.1. Vitamin D and the Innate Immune System

In addition to the expression of VDR in monocytes and macrophages, expression of 1 ∝-hydroxylase, the last enzyme required for the activation of vitamin D, is upregulated in monocytes and macrophages by immunological signaling components, such as signal transducer and activator of transcription-1 (STAT-1∝), interferon-γ (IFN-γ) and Toll-like receptors (TLR) and their ligands (e.g., lipopolysaccharide; LPS) [83,91]. In vitro data also show 1,25-(OH)_2_D3′s anti-inflammatory activity on macrophages, as is demonstrated by its ability to increase concentrations of IL-10 and decrease the concentrations of inflammatory stimuli such as interleukin (IL)-1β, IL-6, tumor necrosis factor (TNF), receptor activator of nuclear factor kappa-B ligand (RANKL) and cyclooxygenase (COX)-2 [92]. 1,25-(OH)_2_D3 inhibits inflammatory cytokines via two pathways. The first involves upregulation of mitogen-activated protein kinase (MAPK)-1 phosphatase by 1,25-(OH)_2_D3, thus resulting in the subsequent inhibition of LPS-induced activation of p38 [93]. The second pathway involves targeting the thioesterase superfamily member 4 to inhibit COX-2 expression. Additionally, in vitro studies have demonstrated that 1,25-(OH)_2_D3 also plays a direct role in the antimicrobial activity of monocytes and macrophages by inducing the expression of cathelicidin antimicrobial peptide (CAMP), consequently resulting in the increased expression of hCAP18 and therefore production of the antimicrobial peptide LL-37 [94,95,96].

1,25-(OH)_2_D3 affects DCs differently from monocytes and macrophages by attenuating DCs towards a less mature and more tolerogenic phenotype, which is accompanied by unique morphological characteristics [88]. DCs are a unique type of antigen-presenting cell (APC) that not only possesses the ability to initiate and direct innate and adaptive immune responses but are also capable of inducing immunological tolerance; the process by which the immune system does not elicit a response against self-antigens [97]. Tolerogenic DCs possess a reduced capacity to process and present antigens and fully activate T cells [97]. Exposure of differentiating DCs to 1,25-(OH)_2_D3 in vitro has been shown to interfere with their differentiation and maturation, consequently locking the cells in a semimature state [98]. Tolerogenic mature DCs treated with 1,25-(OH)_2_D3 are no longer able to activate autoreactive T cells and stimulate the generation of regulatory T cells (Tregs) [99,100,101,102,103].

VDR is also expressed by natural killer (NK) cells and neutrophils [104]. 1,25-(OH)_2_D3 helps to minimize damage to neutrophils caused by pathogens by downregulating inflammatory cytokine production, while simultaneously increasing their effector function against pathogens by increasing the expression of cathelicidin and ∝ and β-defensins [105,106].

#### 2.1.2. Vitamin D and the Adaptive Immune System

1,25-(OH)_2_D3 has been shown to affect T-cells directly and indirectly. 1,25-(OH)_2_D3 indirectly influences T-cells by modulating the stimulatory function of APCs [88]; it does so by downregulating the surface expression of major histocompatibility complex (MHC) class II and co-stimulatory molecules (e.g., CD40, CD80 and CD86) on monocytes and macrophages and DCs, thereby decreasing the potential for antigen presentation [107]. 1,25-(OH)_2_D3 also specifically affects DCs by inhibiting the production of IL-12 and IL-23, cytokines driving differentiation of helper T cells (Th) into Th1 and Th17 phenotypes, respectively, and stimulates the release of anti-inflammatory interleukin (IL)-10 and macrophage inflammatory protein (MIP)-3∝, which recruits CCR4-expressing Tregs [85]. These indirect effects of 1,25-(OH)_2_D3 on monocytes, macrophages and DCs can inhibit the proliferation of autoreactive T-cells, induce early and late apoptosis of autoreactive T-cells, and increase the number of Tregs [100,108]. 1,25(OH)_2_D3 also modulates DC-derived cytokine and chemokine expression by inhibiting the production of pro-inflammatory IL-12 and IL-23, while enhancing the release of anti-inflammatory IL-10 [85,109]. 1,25-(OH)_2_D3′s direct effect on T-cells is variable and dependent upon the activation state of the T-lymphocytes. For example, active T-cells display a higher concentration of VDR, thus allowing 1,25-(OH)_2_D3 to exert a greater effect on these cell [110,111]. 1,25-(OH)_2_D3 has been shown to inhibit the production of Th1-associated cytokines (e.g., IL-2, IFN-γ), Th17 cytokines (e.g., IL-17, IL21) and Th9 cytokines (e.g., IL-9) [109,112,113].

The expression of VDR on B cells also suggests that vitamin D has an influence on their function. In vitro data show that 1,25-(OH)_2_D3 induces apoptosis of activated B-cells and hinders their differentiation to plasma cells and memory B-cells [114]. 1,25-(OH)_2_D3 also upregulates production of IL-10 by B-cells [115]. Finally, 1,25-(OH)_2_D3 downregulates the expression of CD86, and upregulates the expression of CD74, which indirectly reduces the activation of T-cells by dampening the APC function of B cells [92,116].

#### 2.1.3. Vitamin D and Autoimmune Disease

Studies have demonstrated a positive correlation between vitamin D status and risk of autoimmune diseases such as rheumatoid arthritis, multiple sclerosis, and systemic lupus erythematosus (SLE) [117]. In a study conducted by Deluca and Cantorna [87], 1,25-(OH)_2_D3 was found to either prevent or suppress autoimmune encephalomyelitis, SLE, type I diabetes and inflammatory bowel disease, which are all autoimmune diseases characterized by hyperactive T cell-mediated immunity. A possible mechanism by which vitamin D prevents or suppresses autoimmune disorders involving hyperactive T cells is by stimulating the production of transforming growth factor (TGF)-β and IL-4, which suppress T cell- mediated inflammatory responses [87].

Vitamin D is unique among immunoceuticals in that it supports both early immunological activation events and innate host defence mechanisms such as production of defensins, while subsequently also helping to restore immune system homeostasis following activation of adaptive immune responses, by promoting recruitment of Tregs and production of immunosuppressive cytokines.

#### 2.1.4. Vitamin D Dosing

It is important to note that while many countries may have guidelines for recommended daily dietary vitamin D intake, these are designed for optimal bone health rather than for optimal functioning of the immune system. The immune system, however, is generally the most metabolically active in the body, especially when responding to infections or other diseases. Indeed, activated T and B cells can proliferate at rates equivalent to the most rapidly dividing cancer cells. Consequently, requirements for vitamin D could be as high as 60,000 IU/day over a short-term period to ensure optimal immune responses [118].

The current dietary reference intakes (DRI) for vitamin D set out by Health Canada and the US National Institutes of Health (NIH) are solely based on maintaining skeletal health and preventing rickets. Furthermore, Health Canada has confusing guidelines, with one version having a recommended dietary allowance for vitamin D of 600 international units (IU) per day for people 9–70 years of age [119]. The other published guideline recommends 400 IU per day for people 2+ years of age [120].

According to Health Canada’s website on vitamin D and calcium, due to the inconsistent evidence that exists to date and the absence of a cause-and-effect relationship regarding various health outcomes potentially related to calcium and vitamin D, such as cancers, cardiovascular disease, diabetes, and immunity, these health outcomes were not used in determining the updated vitamin D and calcium recommended intake guidelines by the U.S. Institute of Medicine (IOM) [119]. However, because evidence surrounding calcium and vitamin D’s role in bone health was deemed to be convincing by the IOM expert committee, the determination of calcium and vitamin D requirements were based on bone health [119]. Furthermore, the listed tolerable upper intake level (UL) per day by Health Canada and the NIH vastly underestimates the concentration of vitamin D that is considered safe for most individuals. For example, the UL for children and adults from 9–70 years of age set out by Health Canada is 4000 IU. Canadians are also advised that intakes of vitamin D above the new recommended dietary allowance does not offer any additional health benefits, and any intake of vitamin D above the new UL may result in possible adverse effects. Possible adverse effects in this case refer to vitamin D toxicity, which is characterized by an excess amount of calcium being deposited in the body, leading to the calcification of the kidney and other soft tissues including the heart, lungs, and blood vessels. In contrast, numerous studies regarding the use of vitamin D for treating influenza, COVID-19, and pneumonia have consistently shown that vitamin D doses of up to 10,000 IU/day are considered safe for the vast majority of patients [121]. While the IOM also recognizes that no adverse effects have been reported from studies involving the supplementation of less than 10,000 IU/day of vitamin D, the UL is still set at 4000 IU/day due to the presence of U-shaped 25(OH)D concentration-health outcome relationships found in observational studies. However, later investigations have since determined that observational studies reporting J-or U-shaped relationships, did not measure baseline serum 25(OH)D concentrations of participants and thus may have enrolled participants that were already taking vitamin D supplements prior to the study [122]. These U- or J-shaped associations between 25(H)D concentration and health outcomes indicate that the higher the baseline vitamin D status, the greater the risk of adverse outcomes if intake of additional vitamin D for treatment purposes is not adjusted [122].

However, it should be noted that the threshold of toxicity varies from individual to individual as it is highly dependent on lifestyle, diet, skin color, skin type, age, etc. An obese middle-aged individual, for example, with little to no exposure to sunlight, and who consumes a low vitamin D diet will require a much higher dosage of vitamin D to reach toxic concentrations in comparison to a healthy individual in their early 20s with frequent exposure to sunlight and who consumes a balanced diet. In addition to concentration, intake duration of vitamin D must also be taken into consideration when discussing vitamin D toxicity. A single high dose of vitamin D (e.g., 200,000 to 300,000 IU) will not cause adverse effects, as one would need to take daily doses of 25,000 IU of vitamin D for several months or 1 million IU of vitamin D for several days for toxicity to occur [123].

An alternative and more accurate method of determining vitamin D’s safe upper limit is by assessing serum 25(OH)D concentrations (Table 3) [121,124,125,126,127,128].

According to the U.S. IOM 2011 report, which was jointly commissioned and funded by the U.S. and Canadian governments, a serum 25(OH)D concentration of 20 ng/mL (50 nmol/L) or higher is considered adequate for optimal bone health [121]. However, there is increasing evidence that suggests serum 25(OH)D levels of 75 nmol/L and above are sufficient to ensure normal skeletal and muscular structure and function [125,126,129,130]. Furthermore, increasing evidence also suggests that a minimum serum concentration of 100 nmol/L of 25(OH)D is needed to reduce the risk of certain cancers (e.g., colorectal), cardiovascular disease, infectious diseases, pathological pregnancies (e.g., preeclampsia, gestational diabetes, preterm birth), systemic connective tissue diseases, diabetes and COVID-19 [121,123,131,132,133,134,135,136,137,138]. Thus, a serum concentration of at least 100 nmol/L of 25(OH)D appears to be optimal for supporting all systems of the body and not only the skeletal system.

### 2.2. Medicinal Mushrooms and Their Immunomodulatory Properties

Certain mushroom species have long been identified to display profound health promoting benefits. The practice of using mushrooms in Chinese traditional medicine can be dated back to as early as 200–300 AD [139]. Mushroom-producing technologies have greatly advanced to meet the global mushroom market, which was valued at 50.3 billion USD in 2021, and it is expected to expand at a compound annual growth rate of 9.7% from 2022 to 2030 [140]. Examples of medicinal mushrooms include *Lentinula edodes* (Shiitake), *Grifola frondosa* (Maitaki), *Flammulina velutipes* (Enoki), *Pleurotus* (spp) (Osyter), *Ganoderma lucidum* (Reishi), *Trametes versicolor* [139]. *Ganoderma lucidum* and *Trametes versicolor* are not palatable due to their bitter taste and coarse texture, however, they are traditionally used medicinally as hot water extracts [139].

Lentinan is a β-glucan cell wall component extracted from the mushroom *Lentinula edodes* [139,141]. As β-glucans are not found in animals, they can stimulate the immune system by activating various leukocytes including macrophages, DCs, neutrophils, NK cells and lymphocytes [141]. There are several leukocyte surface receptors (e.g., Dectin-1, TLRs, complement receptor type 3, scavenger receptors and lactosyceramide (LacCer)) that recognize β-glucans as non-self molecules, consequently inducing the innate and adaptive immune response [142,143]. Dectin-1 is commonly expressed on neutrophils, DCs, and certain T-cells. The binding of β-glucans to these pattern recognition receptors activates several signaling pathways that promote innate immune responses, including the induction of inflammatory cytokines, activation of phagocytosis and production of reactive oxygen species [141]. The binding of lentinan to TLRs can lead to the production of various cytokines such as IL-2 and IL-12 [141]. The receptor LacCer can be found on neutrophils and endothelial cells, and it is purported that the interaction of lentinan with LacCer will induce production of MIP-2, activation of nuclear factor-κB, and neutrophil oxidative burst [144,145,146].

Polysaccharide krestin (PSK) and polysaccharide peptide (PSP) are protein-bound polysaccharides that are derived from different strains of the mushroom *Trametes versicolor*, also known as Yun Zhi, or turkey tail. PSP and PSK are chemically similar and exert similar physiological effects. The difference between the two lies mainly in the presence of fucose in PSK, and rhamnose and arabinose in PSP [139]. Both PSP and PSK are potent immunostimulators and can increase white blood cell counts, production of IFN-γ and IL-2, and delayed type hypersensitivity reactions [147] suggesting they possess adjuvant-like properties.

There is substantial evidence to suggest that PSP predominantly induces pro-inflammatory cytokines [62]. Both *in vivo* and *in vitro* studies have demonstrated PSP’s potent effect on expression of TNF-∝. For example, in a study conducted by Chan and Yeung [148], the in vitro treatment of mouse peritoneal macrophages with PSP increased the release of TNF-∝ to concentrations comparable after induction by LPS. Cytokines associated with TNF-∝, can also be induced by PSP, as is evident by its induction of IL-12, a Th-1-related cytokine that enhances NK and CD8+ T cell cytotoxic activities and their expression of TNF-∝ [62]. Additionally, the expression of other pleiotropic cytokines, such as TGF-β have been shown to be affected by PSP [62]. Findings also suggests that in addition to PSP’s ability to directly affect cytokine release by leukocytes, PSP also increases the sensitivity of leukocytes to other stimuli, sometimes acting synergistically [62]. Moreover, findings from a study conducted by Li [149] demonstrated that when PSP is added at different concentrations to human blood and mouse splenocytes, it promoted T cell proliferation. Augmentation of Th cell activation, as well as an increase in the ratio of Th cell (CD4+)/T suppressor (CD8+) was also observed following the administration of PSP in the same study.

The administration of PSK under normal physiological conditions has been shown to have no substantial effect on host immune responses [150,151]. However, PSK can help to restore the immune system back to homeostasis following immunological depression caused by tumor burden or chemotherapy [151,152,153]. For example, Harada et al. [154] demonstrated that the oral administration of PSK improved the impaired anti-tumor CD4+ T-cell response in the gut-associated lymphoid tissue of specific pathogen free mice. Kato et al. [155] and Liu et al. [156] also reported that PSK can induce the expression of certain cytokine genes in vivo and in vitro, including TNF-∝, IL-1, IL-8, and IL-6; these cytokines mediate multiple biological effects in part via their direct stimulation of cytotoxic T cells against tumors, enhancement of antibody production by plasma cells and induction of IL-2 receptor expression on T lymphocytes.

In addition to their immunomodulatory properties, the in vitro anticancer activity of both PSP and PSK on human cancer cell lines and in human clinical trials is quite extensive [139]. However, a complete description of their anticancer activities is outside the scope of this review.

### 2.3. Immunomodulatory Properties of Quercetin and Kaempferol

Quercetin and kaempferol are flavonols, a subgroup of flavonoids, a family of polyphenolic compounds that are found abundantly in a variety of fruits and vegetables, including onions, kale, lettuce, tomatoes, apples, grapes, and berries [157], as well as some types of tea [158] and wine [159]. Of note, onions are qualitatively and quantitatively the most important dietary source of quercetin [160], whereas green leafy vegetables such as spinach, kale, dill, chives, and tarragon are the richest plant sources of kaempferol [160].

Quercetin and kaempferol are typically conjugated to simple sugars, such as glucose, xylose, rhamnose, arabinose or galactose or disaccharides (e.g., rutinose) in plants [161,162]. The broad spectrum of health-promoting effects associated with flavonoids make them an essential component of many nutraceutical, pharmaceutical, medicinal and cosmetic applications [157]. Flavonoids are known to possess antioxidative, anti-inflammatory, anti-mutagenic, anti-carcinogenic, and immunomodulatory properties, as well as the ability to modulate key cellular enzyme functions, including the inhibition of the enzymes xanthine oxidase, COX, lipoxygenase (LOX) and phosphoinositide 3-kinase [163,164,165]. For this review, the anti-inflammatory properties of quercetin and kaempferol will be considered as immunomodulatory. The recognition of viruses, bacteria, parasites, antigenic substances and/or chemicals via various cell receptors will activate many inflammatory pathways, resulting in the production of cytokines and activation of leukocyte subsets, such as macrophages and lymphocytes, to eliminate foreign bodies. If the early phase of inflammation fails to eliminate the foreign bodies, an increased production of cytokines, chemokines and inflammatory enzymes may drive inflammation into a chronic phase. The inflammatory pathway is regulated by many receptor-mediated pathways, including TLRs, MAPK pathways and the nuclear factor kappa-light chain enhancer of activated B cells, which a transcription factor that regulates more than 50 genes involved in inflammation. The deregulation of any of these pathways results in the onset and progression of various inflammatory disorders. Quercetin and kaempferol, as well as other flavonoids and polyphenols with anti-inflammatory activities can interact with many molecules involved in the inflammatory pathway to decrease the activity of cytokines, chemokines, and inflammatory enzymes.

Quercetin and kaempferol are known to modulate both innate and adaptive immune responses, exerting stimulatory and inhibitory effects on different types of leukocyte sub-populations and pathways, particularly the inflammatory pathway [166]. The structure of quercetin and kaempferol, and flavonoids in general, is what imparts their anti-inflammatory properties [167]. Quercetin has been shown to reduce inflammation by inhibiting c-Jun N terminal kinase and extracellular signal-regulated kinase, consequently inhibiting MAPK and the transcription factor activator protein -1 and nuclear factor-kB activity [167]. Quercetin is also capable of enhancing the production of IL-10, an anti-inflammatory compound, by inhibiting IL-1β and TNF-∝ [168]. Additionally, quercetin influences the expression of adhesion molecules (e.g., vascular cell adhesion protein 1) and the release of metalloproteinases, thereby reducing inflammation-mediated tissue damage [70]. The anti-inflammatory activities of quercetin are also due its inhibition of inflammation-promoting enzymes, such as COX and LOX, as well as down-regulating nitric oxide (NO) production and/or inducible nitric oxide synthase (iNOS) enzyme expression and activity [169,170], as well as inflammatory mediators [160]. Furthermore, quercetin is known to inhibit the maturation of DCs (derived from murine bone marrow) and their expression of MHC molecules, thereby reducing antigen uptake and the secretion of pro-inflammatory cytokines (IL-1, IL-2, IL-6 and IL-12) [171,172,173,174].Taken together, quercetin modulates immunity and inflammation by mainly acting on DCs and targeting the many intracellular signaling kinases and phosphatases, enzymes and membrane proteins that are often crucial for a specific cellular function [175].

In addition to quercetin’s anti-inflammatory properties, there is also a large body of evidence in the literature to support quercetin’s anti-allergic properties both in vitro and in vivo. Quercetin’s anti-allergic properties, just like its anti-inflammatory properties are directly correlated with its modulation of the immune system. Quercetin is purported to inhibit the production and release of histamine and other allergic and inflammatory substances by stabilizing the cell membranes of mast cells [176,177]. Mast cells play a vital role in the pathogenesis of allergic responses and autoimmune disorders, and they affect the release of many cytokines involved in inflammatory reactions, such as IL-8 and TNF-∝ [178,179]. As such, quercetin is well suited for the treatment of mast cell-derived allergic inflammatory diseases, including asthma, sinusitis, and rheumatoid arthritis [163]. Quercetin is also a known inhibitor of allergic (IgE-mediated) mediator release from mast cells and basophils, and an inhibitor of human mast cell activation [160,180].

Kaempferol, in contrast, exerts its anti-inflammatory activities by inhibiting the nuclear factor-kB binding activity of DNA and myeloid differentiation factor 88, suppressing the release of IL-6, IL-1β, IL-18 and TNF-∝, increasing mRNA and protein expression of Nrf2-regulated genes and inhibiting TLR4 [181,182,183,184]. Like quercetin, kaempferol’s anti-inflammatory properties are also in part due to its inhibition of COX and LOX, pro-inflammatory enzymes, as well as the inhibition of NO production and/or iNOS enzyme expression and activity [168]. Kaempferol also has an immunosuppressive effect on DCs by attenuating their activation [185]. The disruption of nuclear factor- kB and the MAPK pathways, as well as the suppression of calcineurin by kaempferol is one possible mechanism by which kaempferol inhibits DC function [186,187,188,189,190]. Another possible mechanism involves the peroxisome proliferator-activated receptor (PPAR) γ, a transcription factor involved in the anti-inflammatory response, as kaempferol is known to be a potent stimulator of PPARγ activity [190].

### 2.4. Immunomodulatory Properties of Curcumin and Resveratrol

Curcumin is an extract of the rhizome of turmeric (*Curcuma longa* Linn) [191], and its antiangiogenic, antiproliferative, antitumorigenic, antioxidant, and anti-inflammatory properties have been investigated in both in vitro and in vivo studies [192]. Resveratrol [193] is a well-known biologically active compound synthesized by plants and initially isolated from white hellebore (*Veratrum grandiflorum* O. Loes) roots used in traditional Chinese and Japanese Medicine as an anti-inflammatory and anti-platelet agent. Curcumin and resveratrol protect cells from oxidative stress [194], which could have a role in preventing inflammatory disorders [195], such as cancer, inflammation, and diabetes. Curcumin and resveratrol have been widely reported to have anticancer properties [192,196], and evidence suggests the significance of these phytochemicals as preventive and therapeutic agents that can effectively target CRC development and progression [197]. However, it has been documented that these polyphenols exhibited immunosuppression activities by the down-regulation of costimulatory molecule expression, CD28, and CD80; up-regulating CTLA-4 on macrophages; and the augmentation of the IL-10 generation. Both compounds suppress the activity of T and B cells, as evidenced by significant inhibition in proliferation, antibody production, and lymphokine secretion [198]. Further studies are required to dissect these phytochemicals’ chemopreventive effects and their anticancer mechanisms.

Cellular inflammatory mediator secretion is under the regulation of different signaling pathways, including the IκB kinase β (IKKβ) and NF-κB pathways [199]. It has been shown that curcumin and resveratrol suppress NF-κB-regulated gene products involved in osteoarthritis, and IL-1β-induced NF-κB activation was shown to be suppressed directly by cocktails of curcumin and resveratrol [200]. We have also reported that treating human macrophages with curcumin inhibits IL-8 production by 85% (from ~52.5 ± 7 ng/mL to ~4.2 ± 0.3 ng/mL) via the suppression of NF-κB upon cigarette smoke exposure [201]. However, studies show that supplementation with curcumin and resveratrol has no impact on the postprandial inflammation response to a high-fat meal in abdominally obese older adults [199], demonstrating that the whole-blood *NFKB1* gene expression and C-reactive protein, as well as the generation of inflammatory cytokines IL-6 and IL-8, stayed unchanged. Conversely, the oral administration of resveratrol and curcumin ameliorates acute small intestinal inflammation by down-regulating Th1-type immune responses and prevents bacterial translocation by maintaining gut barrier function [202]. Other studies [203] have shown that curcumin and resveratrol could regulate weaned piglet gut microbiota, down-regulate the TLR4 signaling pathway, alleviate intestinal inflammation, and ultimately increase intestinal immune function. These findings indicate that the gut microbial metabolites of the polyphenol supplementation, which contribute to the clinical effects, need to be investigated.

Considering all the research conducted so far, there is no doubt that curcumin and resveratrol are preventive/therapeutic agents that can be used in the management and treatment of cancer and inflammatory diseases. However, further investigation should clarify how these classes of dietary micronutrients should be administered to maximize their anti-carcinogenic, anti-angiogenic, pro-apoptotic, or anti-oxidative effects and enhance their beneficial effects in the prevention or control of several diseases.

## 3. Conclusions

There is a significant and growing need for innovative and alternative treatment options as well as novel healthful products to prevent and/or treat the increasing number of infectious diseases, cancers, autoimmune disorders, inflammatory conditions, and allergies affecting both human and animal populations in a more natural manner. As the nutraceutical market expands, it is becoming increasingly clear that more studies are needed to potentially establish new definitions of sufficiency and deficiency based on immunological health parameters, as well as for the defining and testing of immunocompetence. Currently, all recommended nutrient intake guidelines are based on preventing deficiency. With regard to vitamin D, for example, the current recommended intake guidelines set out by Health Canada and the FDA are solely based on maintaining skeletal health and preventing rickets and have nothing to do with optimizing the function of the most metabolically active system in the body, the immune system. Indeed, the immune system requires a much higher vitamin D intake (e.g., as much as 4000–10,000 IU versus 400–600 IU). Furthermore, the concentration of vitamin D required for an optimal immunological function for one individual differs from another due to lifestyle (e.g., use of sunscreen and clothing choice), level of skin pigmentation, exposure to the sun, nutritional habits, and genetics, as well as the season and geographical latitude of residence. As such, there is a need for high-quality studies to assess the roles of vitamin D in optimizing immunological functions, especially in clinically relevant contexts and with precise measurements of serum concentrations. Defining and standardizing testing for immunocompetency would allow for a more focused, personalized approach to supporting the immune system and ensuring its optimal functioning to minimize the acquisition of diseases.

Additionally, as the list of potential immunoceuticals expands, additional studies are required to determine their mechanisms of action on the immune system, the concentrations at which they exert their immunomodulatory properties, optimal methods of delivery, how well in vitro studies translate into in vivo results, and whether they can serve as stand-alone treatment options or act as adjunctive therapies for certain pathological conditions and disorders. Although standardized methods, such as the High Immune Response technology, are available for livestock, more studies are required to determine if certain immunoceuticals can be used in various animal species to help treat, prevent, or augment the treatment of common and highly infectious diseases afflicting the livestock industry, including respiratory diseases in swine, dairy, beef, and poultry, as well as mastitis, weaning disorders, and parasitic infections, to name a few. The immunoceuticals industry is full of potential to provide natural, economical, and efficacious alternatives to traditional pharmaceuticals and fill therapeutic gaps that currently exist in the drug industry. In short, they have the potential to revolutionize how pathological conditions are treated or prevented through the modulation of the immune system.

## Figures and Tables

**Figure 1 nutrients-14-04075-f001:**
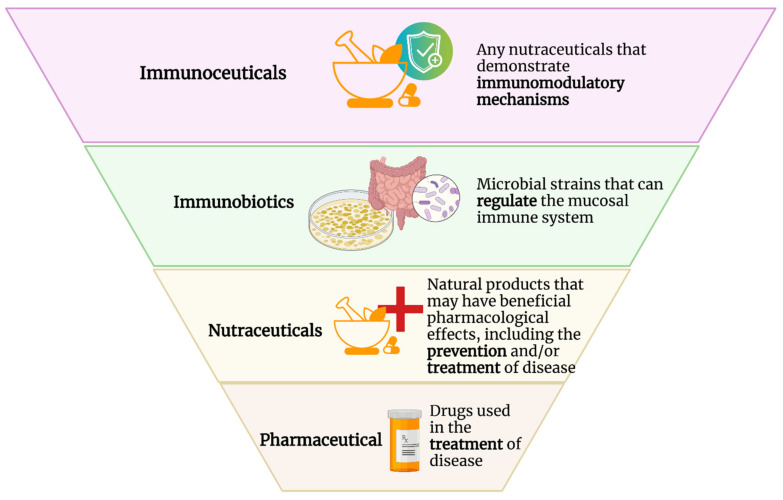
Comparison of the terms “pharmaceutical”, “nutraceutical”, “immunobiotics”, and “immunoceuticals”.

**Figure 2 nutrients-14-04075-f002:**
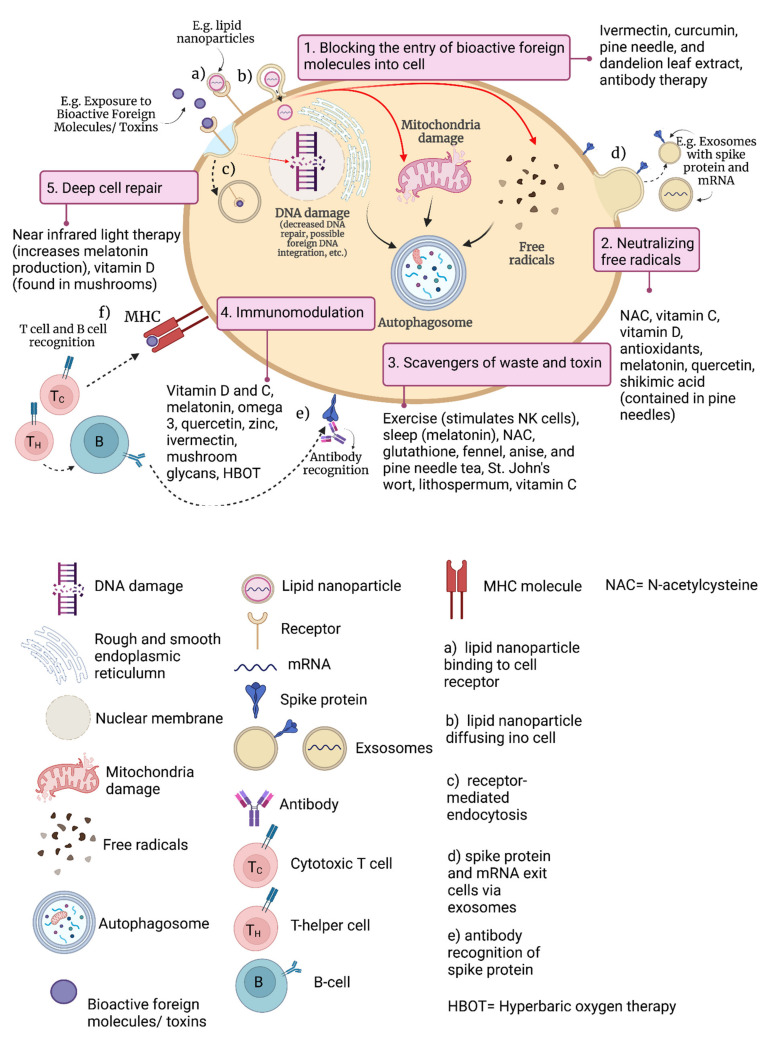
Ways to naturally optimize and regulate the immune system (e.g., following infection with severe acute respiratory syndrome-coronavirus-2 (SARS-CoV-2), or an adverse event following vaccination against SARS-CoV-2, both resulting in exposure to spike protein). MHC = major histocompatibility complex.

**Figure 3 nutrients-14-04075-f003:**
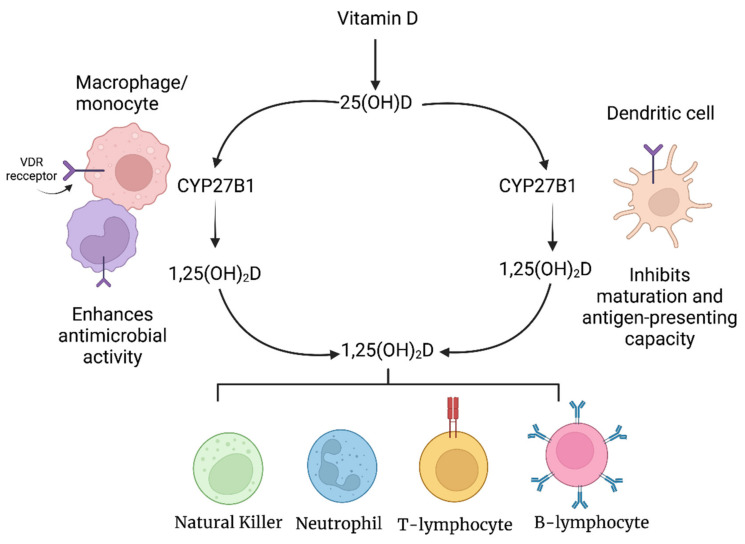
Effects of vitamin D on key cells of the immune system.

**Table 1 nutrients-14-04075-t001:** Clinically proven immunoceuticals for treating respiratory and/or gastrointestinal tract infections, cancers, and acquired immunodeficiency syndrome (AIDS).

Immunoceuticals	Authors	Study Design	Disease/Pathological Condition Addressed with Immunoceuticals	Dose	Results
Vitamin D3 (Cholecalciferol)	[31]	Multi-center, randomized clinical trial	Severe acute respiratory syndrome- coronavirus-2 (SARS-CoV-2) infection	5000 IU or 1000 IU of Vit D3 once daily for two weeks	Vit D supplementation significantly increased serum 25(OH)D levels in the 5000 international units (IU) group 5000 IU of daily Vit D3 supplementation reduced recovery time for cough and gustatory sensory loss in patients with mild to moderate COVID-19
	[32]	Quasi-experimental study	SARS-CoV-2 infection	Oral bolus of 80,000 IU Vit D3 during or just before infection with COVID-19	82.5% of participants in intervention group survived infection with COVID-19 versus 44.4% in comparator group Intervention group had longer survival time than Comparator group (log-rank *p* = 0.002) Vit D3 supplementation inversely associated with Ordinal Scale for Clinical Improvement score for COVID-19 (β = −3.84 (95% CI:−6.07;−1.62), *p* = 0.001)
	[33]	Randomized clinical trial	SARS-CoV-2 infection	Oral supplementation of 10,000 IU daily of Vit D3 for 14 days	10,000 IU of Vit D3 daily for 14 days was sufficient to raise Vit D concentrations supplemented group presented fewer symptoms than non-supplemented on day seven and fourteen of follow-up
	[34]	Multicenter, randomized, double-blind, placebo-controlled, parallel-group trial	Influenza A infection	1200 IU of Vit D3 daily	Vit D3 supplementation during winter season may reduce incidence of influenza A-mediated illness Influenza A infections occurred in 18/167 children in the Vit D3 group versus 31/167 children in the placebo group
Polysaccharide K (PSK)	[35]	Randomized double-blind trial	Colorectal cancer	3 g/day starting 10–15 days after surgery until two months after surgery, then 2 g daily until 24 months and 1 g daily thereafter	rate of patients in remission was significantly higher in PSK group versus placebo group Survival rate in PSK group significantly higher (*p* < 0.05) than control group Polymorphonuclear leukocyte activities in PSK-treated patients significantly enhanced
	[36]	Randomized, controlled trial	Colorectal cancer	3 g of PSK per day for over three years	Disease-free and overall survival of PSK group were longer than those of the control group
Polysaccharide-Peptides (PSP)	[37]	Double-blind placebo-controlled randomized study	Non-small cell lung cancer (NSCLC)	340 mg of purified Yun-zhi PSP capsules three times daily for 28 days	significant improvement in blood leukocyte and neutrophil counts, serum IgG and IgM, and % of body fat in PSP group, but not control (*p* < 0.05) 5.9% of PSP patients were withdrawn due to disease progression versus 23.5% of control patients PSP treatment associated with slower deterioration in advanced NSCLC patients
Probiotics	[38]	Randomized controlled open-label trial	Acute upper respiratory tract infections (acute URTI)	300 mL/day of yogurt supplemented with a probiotic strain, *Lactobacillus paracasei* N1115 (N1115), 3.6 × 10^7^ CFU/mL for 12 weeks	number of persons diagnosed with acute URTI and number of URTI events significantly decreased in probiotic group versus control Percentage of CD3^+^ cells in intervention group significantly higher than in control
Prebiotic and probiotic	[39]	Community based double-masked, randomized controlled trial	Diarrhea, respiratory infections and severe illnesses in children aged 1–4 years of age	milk fortified with 2.4 g/day of prebiotic oligosaccharide and 1.9 × 10^7^ CFU of probiotic *Bifidobacterium lactis* HN019	incidence of dysentery episodes, pneumonia and severe acute lower respiratory infection reduced by 21%, 24%, and 35%, respectively
Quercetin	[40]	Prospective, randomized, controlled, and open-label study	SARS-CoV-2 infection	1000 mg of Quercetin/day for 30 days	statistical improvement of all clinical outcomes (need and length of hospitalization, need of non-invasive oxygen therapy, progression to Intensive care unit, and number of deaths) 1000 mg of Quercetin/day was well tolerated by all subjects
	[41]	Second, pilot, randomized, controlled and open-label clinical trial	SARS-CoV-2 infection	600 mg of Quercetin/day for seven days, followed by 400 mg of Quercetin/day for another seven days	16 of the 21 COVID-19 outpatients in the Quercetin group tested negative for SARS-CoV-2, and 12 patients in the Quercetin group had all their symptoms diminished one-week post-treatment Quercetin significantly improved virus clearance, symptom frequency, lactate dehydrogenase, and ferritin
Beta-Carotene (Carotenoid)	[42]	Pilot study	AIDS	60 mg/day for four weeks	Total lymphocyte counts increased by 66% and CD4+ cells rose slightly Patients with a baseline CD4+ cells greater than 10/ul demonstrated an average increase of 53 ± 10 cells/ul
Omega-3 fatty acids	[43]	Single-blind randomized controlled trial	SARS-CoV-2 infection	2 g of docosahexaenoic acid (DHA) + eicosapentaenoic acid (EPA) for 2 weeks	significantly decreased fatigue and body pain, and increased appetite in intervention group decreased erythrocyte sedimentation rate and C-reactive protein following two weeks of omega-3 supplementation
	[44]	Double-blind, randomized clinical trial	SARS-CoV-2 infection	One capsule of 1000 mg omega-3 daily containing 400 mg EPAs and 200 mg DHAs for 14 days	significantly higher one-month survival rate and higher arterial pH, HCO_3_, and base excess (respiratory function parameters) and lower blood urea nitrogen, creatinine, and potassium (renal function parameters) in intervention group versus control group (both composed of critically ill COVID-19 patients)
Melatonin	[45]	Single-center, double-blind, randomized clinical trial	SARS-CoV-2 infection	3 mg of melatonin three times daily for 14 days	Significant improvement in clinical signs and symptoms (cough, dyspnea and fatigue), as well as C-reactive protein concentrations and pulmonary involvement in intervention versus control Significantly shorter mean time of hospital discharge and return to baseline health in intervention versus control

COVID-19 = Novel coronavirus disease identified in 2019; SARS-CoV-2 = severe acute respiratory syndrome-coronavirus-2; I.U. = international units; PSK = polysaccharide K; PSP = polysaccharide-peptides; Vit D3 = vitamin D3/cholecalciferol; CFU = colony forming units; URTI = upper respiratory tract infection; CD = cluster of differentiation; DHA = docosahexaenoic acid; EPA = eicosapentaenoic acid; NSCLC = non-small cell lung cancer; IgG = immunoglobulin G; IgM = immunoglobulin M; HC0_3_ = bicarbonate.

**Table 2 nutrients-14-04075-t002:** Immunomodulatory properties of immunoceuticals.

Immunoceutical	Immunomodulatory Properties
Vitamin D3 (Cholecalciferol)	Regulates production of antimicrobial peptides (cathelicidin and defensin); ↑ expression of antimicrobial peptides [56] ↑ expression of proteins involved in intercellular connections (connexin-43, tight junctions, and E-cadherin) in epithelial barriers [56] Vitamin D receptor (VDR) expressed in almost all leukocytes (i.e., activated CD4^+^ and CD8^+^ T cells, B cells, and antigen-presenting cells, such as macrophages and dendritic cells); receptor-ligand pair (Vit D3 and VDR) acts as a strong immunosuppressor [56,57] Enhances mobility and phagocytosis of macrophages, and ↑ generation of tumor necrosis factor (TNF)-∝ by macrophages [58] Causes neutrophils to traffic to sites of inflammation and stimulates them to kill microbes [56] Controls interferon (IFN) production [56] Inhibits the proliferation, differentiation, and production of antibodies by B cells [56] Inhibits differentiation and maturation of dendritic cells, ↓ expression of major histocompatibility complex (MHC)-II and auxiliary stimulative molecules such as B7 and CD40 on dendritic cells, and thus, ↓ cytotoxicity of CD8^+^ T cells [56] Reduces T lymphocyte proliferation and regulates skewing towards particular CD4^+^ T cell subsets [57] Shifts cytokine patterns from a Th-1 to a Th-2 milieu by inhibiting cytokines required for Th1 differentiation (e.g., IL-12) or produced by differentiated Th1 cells (e.g., IL-2 and IFN-*γ*), and augmenting Th2 cell development to promote self-tolerance [57,59] Activates renin-angiotensin system (RAS) pathway by inducing transforming growth factor (TGF)-*β*-1 [56] Reduces risk of developing autoimmune diseases (e.g., Type 1 diabetes mellitus, multiple sclerosis, rheumatoid arthritis, systemic lupus erythematosus, Crohn’s disease, thyroiditis, psoriasis, polymyalgia rheumatic, autoimmune gastritis, and systemic sclerosis) [60,61]
Coriolus versicolor extract (polysaccharide krestin (PSK), polysaccharide peptide (PSP))	PSP induces expression of TNF-∝; pro-inflammatory cytokine with potent tumoricidal activity and capable of inducing apoptosis [62] PSP increases production of IL-12; Th1-related cytokine that enhances the cytotoxic activities of natural killer and CD8^+^ T cells and their expression of TNF-∝ [62] PSP induces interleukin (IL)-1*β*; a pleiotropic cytokine and pro-inflammatory signal to enhance lymphocyte proliferation and differentiation [63,62] Coriolus versicolor (CV) extract activates T lymphocytes, B lymphocytes, monocytes/macrophages, bone marrow cells, natural killer cells and lymphocyte activated killer cells in vitro [64] Promotes the proliferation and/or production of antibodies and various cytokines (i.e., IL-2 and IL-6, IFNs, and TNF-∝) [64] CV extracts shown to restore certain depressed immunological responses caused by tumor burden or chemotherapy treatment to normal levels [65,66,67]
Quercetin	Neutralizes free radicals through the donation of hydrogen atoms; antioxidant activity increases cell survival rate [68] Strong reducing agent; provides protection against oxidative stress [69] Inhibits production of cyclooxygenase (COX) and lipoxygenase (LOX) inflammatory enzymes [70] Induces gene expression and production of Th-1 derived IFN-*γ*, and down-regulates Th-2-derived IL-4 by blood mononuclear cells [70] Direct regulatory effect on basic functional properties of leukocytes via the extracellular regulated kinase 2 (Erk2) mitogen-activated protein kinase (MAPK) signaling pathway in human mitogen-activated blood mononuclear cells and purified T lymphocytes [70]
Carotenoids	Strong antioxidant agents to combat oxidative stress caused by cytokine storm induced by the innate immune system in response to viral infections [71] Vitamin C improves pulmonary function and decreases risk of acute respiratory distress syndrome [71] Vitamin E alleviates oxidative damage and inflammation induced by SARS-CoV-2 [71] *β*-carotene and lycopene possess anti-inflammatory properties due to their reactive oxygen species (ROS)-scavenging activities [71]
Probiotics	Activates naïve T and B cells: probiotics and their antigenic metabolites can be phagocytosed by microfold cells forming endosomes that can be released and acquired by dendritic cells, which then transports them to local lymph nodes [72] Induces the release of antimicrobial defensins from epithelial cells [72] Modulates innate and adaptive immune responses, and facilitates the development and maturation of the immune system [72] Regulates host-pathogen interactions by initiating innate immune responses; composed of Toll-like receptors, nuclear factor kappa B (NF-κB), MAPK, and c-Jun NH2-terminal kinase (JNK) pathways [72] Enhances viability of natural killer cells and macrophages [72] Stimulates release of secretory IgA [72]
Omega-3 fatty acids	Upregulates the activation status of macrophages, neutrophils, T-cells, B-cells, dendritic cells, natural killer cells, mast cells, basophils, and eosinophils [73] Modulates neutrophil function via neutrophil migration, phagocytic capacity, and production of ROS and cytokines [74] Activates function of T cells by promoting antigen-presenting cells (APC) [73] Improves function of macrophages by secreting cytokines and chemokines, promoting phagocytosis, and activating macrophages via polarization [73] Downregulates NF-κB [73] Anti-inflammatory due to production of different prostaglandins, lipoxins, and peroxisome proliferator-activated receptor gamma (PPAR*γ*) [73] Affects cell signaling by affecting lipid raft formation and functions [73]
Melatonin	Potent free radical scavenger and antioxidant; detoxifies various ROS and reactive nitrogen species [75] Stimulates the activities of several antioxidant enzymes and/or upregulates their gene expression [75] Suppresses activity or downregulates gene expressions of several proinflammatory enzymes (e.g., COX2, inducible nitric oxide synthase (iNOS), eosinophilic peroxidase, and matrix metallopeptidase 2 and 9 (MMP2,9)) [75] Suppresses NLRP3 inflammasome progression [75] Inhibition of IκBα phosphorylation suppresses the cytokine storm [75] Downregulates the overreaction of the innate immune response and promotes adaptive immunity [75] Inhibits migration of neutrophils to inflammatory sites by blocking ERK phosphorylation [75] Downregulates mast cell activation, ↓ production of TNF-∝ and IL-6, and inhibits IKK/NF- κB signal transduction pathway in activated mast cells [75] Balances ratio of T lymphocyte subpopulations and ↑ numbers of B lymphocyte and antibody titer following vaccination [75]

Vit D3 = vitamin D3/cholecalciferol; VDR = vitamin D receptor; CD = cluster of differentiation; TNF-∝ = tumour necrosis factor alpha; IFN = interferon; IFN-*γ* = interferon gamma; IL = interleukin; MHC = major histocompatibility complex; TGF-*β* = transforming growth factor beta; RAS = rennin angiotensin system; PSK = polysaccharide K; PSP = polysaccharide-peptides; CV = coriolus versicolor; COX = cyclooxygenase; LOX = lipoxygenase; ERK2 = extracellular regulated kinase 2 MAPK = mitogen-activated protein kinase; SARS-CoV-2 = severe acute respiratory syndrome-coronavirus-2; Th = T helper cell; APC = antigen presenting cell; ROS = reactive oxygen species; IKKβ = IκB kinase β; NF-κB = nuclear factor kappa B; JNK = c-Jun NH2-terminal kinase; IgA = immunoglobulin A; PPAR*γ* = peroxisome proliferator-activated receptor gamma; MMP = matrix metallopeptidase; iNOS = inducible nitric oxide synthase. ↑ = increased; ↓ = decreased.

**Table 3 nutrients-14-04075-t003:** Vitamin D status categorized by serum 25(OH)D concentrations.

Vitamin D Status	Serum 25(OH)D Concentrations (nmol/L)	Serum 25(OH)D Concentrations (ng/mL)
Severe deficiency	<25	<10
Moderate deficiency	25–<50	10–<20
Insufficiency	50–<75	20–<30
Sufficiency	75–<100	30–<40
Optimal concentration	100–<150	40–<60
Increased concentration	150–<250	60–<100
Overdose	≥250	≥100
Intoxication	≥375	≥150

## Data Availability

Not applicable.

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
