# Peer review of "Immunoceuticals: Harnessing Their Immunomodulatory Potential to Promote Health and Wellness"

_nutrients, 2022, doi:10.3390/nu14194075_

Round 1
Reviewer 1 Report
The manuscript entitled “Immunoceuticals: Harnessing their Immunomodulatory Potential to Promote Health and Wellness” extensively reviewed nutraceuticals with immunomodulatory properties. In the first step, the in-depth detail about importance of immunomodulation and testing immunocompetence has been provided. The second-half of the manuscript provides the overview of immunoceuticals (nutraceuticals with immunomodulation capabilities). The work is novel and in-depth insight to clinically valuable immunoceuticals. Overall, the manuscript is well written.
However, few queries needs to be answered before publication.
1. Graphical abstract must be provided
2. Several nutraceuticals with immunomodulatory properties are missing for example decades of research studies have shown that curcumin and resveratrol possess immunomodulatory properties. These must be included
Author Response
"Please see the attachment."
Please note the graphical abstract will be uploaded later, recommended by the editor assistant.

Reviewer 2 Report
-Some parts in the manuscript contain short sentences, try to conjugate also this section lacks references, however I agree that there are some parts do not appear to be taken from specific sources. See Line 76-102: Short sentences,
- Line 175: “As such, developing anti-inflammatory therapeutic interventions, including immunoceuticals, that are safe and effective remains a crucial goal” Why do not you use “As such, developing safe and effective anti-inflammatory therapeutic interventions, including immunoceuticals, remains a crucial goal” instead, to avoid confusion?.
- Line: 189: “It has been well-established in animal models..” Good mention but Can you refer to human studies?
- Why do not you refer to the pandemic (COVID-19)? I think this article can help: https://pubmed.ncbi.nlm.nih.gov/34061781/
- In table 1 try to focus on results of immunity
- Some little typo errors are seen (eg in Line 552)
- Conclusion includes precious home messages, however if you can concise would be better.
- Reference should be trimmed, example Line 804 “(accessed on 27 August 2022 “ should be removed
Author Response
"Please see the attachment."

Reviewer 3 Report
The authors should classify the immunoceuticals based on that different classes and their mechanisms should be discussed.
In current form the literature is very generally discussed, authors should also provide their input what are the reasons and cellular mechanisms involved.
Immunoceuticals is a very broad term and should incorporate all possible groups within the rreview.
How data was collected to construct the review should also be discussed within the review article.
Author Response
"Please see the attachment."

Round 2
Reviewer 3 Report
The authors have not responded to previously raised concerns, a general overview and talking about few examples as a representative of immunoceuticals do not justify the manuscript as review article.